# The Expression of Proto-Oncogene *ETS-Related Gene* (*ERG*) Plays a Central Role in the Oncogenic Mechanism Involved in the Development and Progression of Prostate Cancer

**DOI:** 10.3390/ijms23094772

**Published:** 2022-04-26

**Authors:** Ealia Khosh Kish, Muhammad Choudhry, Yaser Gamallat, Sabrina Marsha Buharideen, Dhananjaya D, Tarek A. Bismar

**Affiliations:** 1Department of Pathology and Laboratory Medicine, Cumming School of Medicine, University of Calgary, Calgary, AB T2V 1P9, Canada; ealia.khoshkish@ucalgary.ca (E.K.K.); muhammad.choudhry@ucalgary.ca (M.C.); yaser.gamallat@ucalgary.ca (Y.G.); sabrina.buharideen@ucalgary.ca (S.M.B.); dhananjayad@ucalgary.ca (D.D.); 2Alberta Precision Laboratories, Calgary, AB T2V 1P9, Canada; 3Departments of Oncology, Biochemistry and Molecular Biology, Calgary, AB T2V 1P9, Canada; 4Tom Baker Cancer Center, Arnie Charbonneau Cancer Institute, Calgary, AB T2V 1P9, Canada

**Keywords:** *ETS-related gene* (*ERG*), *TMPRSS2-ERG* gene fusion, prostate cancer, tumorigenesis

## Abstract

The *ETS-related gene* (*ERG*) is proto-oncogene that is classified as a member of the *ETS* transcription factor family, which has been found to be consistently overexpressed in about half of the patients with clinically significant prostate cancer (PCa). The overexpression of *ERG* can mostly be attributed to the fusion of the *ERG* and *transmembrane serine protease 2* (*TMPRSS2*) genes, and this fusion is estimated to represent about 85% of all gene fusions observed in prostate cancer. Clinically, individuals with *ERG* gene fusion are mostly documented to have advanced tumor stages, increased mortality, and higher rates of metastasis in non-surgical cohorts. In the current review, we elucidate ERG’s molecular interaction with downstream genes and the pathways associated with PCa. Studies have documented that *ERG* plays a central role in PCa progression due to its ability to enhance tumor growth by promoting inflammatory and angiogenic responses. *ERG* has also been implicated in the epithelial–mesenchymal transition (EMT) in PCa cells, which increases the ability of cancer cells to metastasize. In vivo, research has demonstrated that higher levels of ERG expression are involved with nuclear pleomorphism that prompts hyperplasia and the loss of cell polarity.

## 1. Background

In the past four years, the annual diagnostic rate of PCa cases have risen from 161,360 to 248,530 patients in the US alone, which is indicative of a 54% increase. Consequently, prostate cancer (PCa) has become increasingly prevalent within men, especially in western populations [1]. Currently, PCa is now the most common type of cancer and is the second leading cause of death among male cancer patients [2,3].

The *ETS-related gene* (*ERG*) is a transcription factor that is encoded by *ERG* and has been identified to be consistently overexpressed in the malignant epithelial cells found in PCa, demonstrating its potential as a biomarker or therapeutic target [4,5].

*ERG* was first discovered in 1987 when a cDNA clone was isolated from a library of colon cancer cells and was labeled as a new member of the E-26 transformation-specific (*ETS*) oncogene family [5]. *ETS* and *ETS*-related genes have been demonstrated to behave as major regulators of transcriptional activity and are directly involved in proliferation, differentiation, vasculogenesis, and angiogenesis [6,7]. Bioinformatic analysis has uncovered various fusions in the 5’ untranslated region of *TMPRSS2* (21q22) when it is fused to *ERG* (21q22), *ETV1* (7p21), *ETV4* (17q21), or *ETV5* (3q27), resulting in the overexpression of ETS and ETS-related genes in prostate cancer [8,9,10,11]. *TMPRSS2*-*ERG* fusion appears in approximately 50% of PCa peripheral zone tumors, while this fusion is only apparent in 12% of PCa transition zone tumors [12,13]. Additionally, *ERG* rearrangements seem to occur at variable frequencies depending on the studied population, with Caucasian Americans expressing *ERG* at a frequency of 50–55%, and African Americans showing an *ERG* frequency of 28% [14]. It was also observed that populations of Asian descent had varying *ERG* frequencies, ranging from 23% in Chinese PCa patients to 49% in Indian PCa patients [15]. We illustrate the fusion mechanism and its prevalence among different ethnicities in Figure 1.

Under normal physiological conditions, *ERG* has many developmental functions that vary depending on the cell type or the organism’s developmental stage. During embryogenesis, ERG has been shown to be highly expressed in the mesoderm and endothelium, where it plays an important role in vasculogenisis and in the development of bone [16]. Other endothelial genes that are positively regulated by *ERG* include vascular endothelial growth factor (*VEGF*), the von Willebrand factor, and Endoglin, which are all involved in endothelial cell differentiation and angiogenesis [17,18].

It appears that *TMPRSS2-ERG* gene fusion plays a role in inhibiting androgen receptor-driven differentiation, therefore producing de-differentiated cells with stem-like properties that are correlated with the EMT pathway [19,20]. The overexpression of *ERG* has also been observed in other cancer subtypes, such as leukemia and Ewing’s sarcoma, which indicate its potent potential as an oncogene [21]. In vivo, it has been demonstrated that the overexpression of *ERG* induces neoplastic changes, and its overexpression has also been correlated with epithelial nuclear pleomorphism and the loss of cell polarity [21,22]. Clinically, individuals with higher *ERG* expression have been found to have more advanced tumor stages, elevated Gleason scores, increased mortality, and metastasis. The overexpression of *ERG* has consistently been identified in the malignant epithelial cells found in PCa, demonstrating its potential as a possible biomarker and therapeutic target [23]. In this study, we reviewed and elucidated the oncogenic role of *TMPRSS2-ERG* fusion in PCa and the mechanism underlining its association with PCa progression and metastasis.

## 2. Structural Characteristics and Allosteric Autoinhibition of *ERG*

The ERG transcription factor has a primary structure that consists of 486 amino acids and a corresponding molecular weight of 54kDa [21]. A distinguishing characteristic of ETS family proteins is the presence of a DNA-binding domain called the ETS DNA binding domain (EBD). This EBD domain is made of 85 amino acids and contains 3 alpha-helices that are further supported by a 4-stranded anti-parallel beta sheet [24]. The EBD domain plays a critical role in DNA recognition as well as in AP-1 and co-activator recruitment [25]. Within the EBD, there are three highly conserved tryptophan residues that serve as a hydrophobic core to facilitate the helix–turn–helix binding domain in proteins [26]. ERG analysis by means peptide sequencing has also predicted the presence of phosphorylation sites for protein kinase C and a pointed (PNT) domain in the N-terminus [21]. The PNT domains are a part of a larger sterile alpha motif (SAM) family that is involved in many diverse protein–protein interactions that can allow for self-association [27]. This PNT domain has also been shown to facilitate the heterodimerization of ERG with other proteins, including other members of the ETS family, DNA-dependent kinases, AP-1 complex, and the androgen receptor (*AR*) [28].

## 3. The Function of *ETS-Related Gene* (*ERG*) in Normal Cell Types

ERG has a multitude of physiological functions that differ based on the type of cells or the organism’s developmental stage. During embryogenesis, ERG has been observed to be highly expressed in the mesoderm and endothelium, where it plays a crucial role in vasculogenesis and in the development of bone [29]. In adults, it regulates vascular homeostasis and angiogenesis by activating the transcription of endothelial specific genes, such as vascular endothelial (VE)-cadherin, an adhesion molecule that promotes vascular stability by maintaining and controlling endothelial cell contact [30]. In addition to this, VE-cadherin also plays a central role in cell proliferation and apoptosis and modulates endothelial growth factor receptor functions [31]. Other endothelial genes that are positively regulated by ERG include the vascular endothelial growth factor (*VEGF*), von Willebrand factor, and endoglin, which are all involved in endothelial cell differentiation and angiogenesis [26,32].

## 4. Prominent *TMPRSS2-ERG* Gene Fusion Found in PCa

In prostate cancer cells, a surprisingly common occurrence involves the fusion of *ERG* to *TMPRSS2*, which forms the fusion product of *TMPRSS2-ERG*. The most common mechanism by which these two genes fuse involves the deletion of intronic sequences on the long arm of chromosome 21 via an intron deletion between TMPRSS2 and ERG on chromosome 21q22.2-3 (Figure 1). This fusion mechanism has been identified as being prevalent in approximately 50% of prostate cancer patients [33]. The frequent occurrence of this fusion protein can be attributed to the presence of a homogenous deletion site that is present between *ERG* and *TMPRSS2* [34]. Moreover, this deletion site is separated into two different classifications according to various start sites. In both of the deletion products, the 5′ end of the *TMPRSS2* gene has been ligated to the 3′ end of *ERG*. *TMPRSS2-ERG* fusion results in *ERG* overexpression due to the androgen responsive promoter of the *TMPSS2* gene allowing for the constitutive transcription of *ERG,* which has been shown to be correlated with increased cell proliferation, cell invasion, angiogenesis, and invasiveness in PCa cells [35,36]. In addition, this *TMPRSS2-ERG* fusion enhances the transcription and activates downstream oncogenes [37].

## 5. Functional *ERG* Overexpression in Prostate Cancer Cells

In prostate cancer cells, *ERG* overexpression increases the rate of epithelial to mesenchymal transitions via the EMT pathway, enhancing the ability of PCa cells to invade and metastasize. *ERG* achieves this by upregulating matrix metalloproteinases (*MMPs*), *CXCR4*, and *Osteopontin* (*OPN*), which have been correlated with higher rates of cell invasion and metastasis among patients [38]. Additionally, the *TMPRSS2*-*ERG* pathway reveals the epithelial to mesenchymal transition via the *ZEB1/ZEB2* axis in PCa [39]. The constitutive expression of ERG also hyperactivates the inflammatory pathway in PCa cells by binding to *Toll-like receptor 4*; this activates the NF-kb pathway, increasing the transcription of target genes such as *TNFA*, *IL6*, *BCLXL*, *BCL2*, *BCLXS*, *XIAP*, and *VEGF* [40,41]. These proteins trigger tumor growth and progression by enhancing cell proliferation, survival, and angiogenesis. Tumor growth is further accentuated by the activation of the *EZH2* promoter by *ERG.* This relieves the epigenetic inhibition of tumor suppressor genes such as *NKX3.1*, resulting in the constitutive expression of the *TMPRSS2-ERG* fusion gene. In addition to its role in regulating tumor cell invasion and proliferation, *ERG* also plays an important role in negatively regulating tumor cell differentiation by inhibiting the transcription of genes such as *KLK3*/*PSA* and *SLC45A3*/*Prostein* [42]. Altogether, the overexpression of *ERG* in prostate cancer is a key modulator of tumor progression and aggressiveness, as it can regulate the transcription of the proteins that mediate inflammation, cell invasion, differentiation, and oncogenesis [43].

## 6. *ERG* Ameliorates Cell Cycle Driving Genes

In *ERG* knockout mice, there was significant decrease in the number of cells arrested at G0 and an increase in cells at G_1_. This demonstrates ERG’s role in maintaining the homeostasis of the hematopoietic stem cell (HSC) population. This suggests that ERG is a major cell cycle regulator in HSC niches and that it works to maintain a constant balance between cell differentiation and renewal. It is revelated that in the absence of *Phosphate and Tensin Homolog* (*PTEN)*/*TP53*, ERG binds directly to the chromatin loci of various cell cycle-driving genes and reduces their expression while activating the RB gene [44,45]. This leads to *E2F1* inhibition and, interestingly, the stability of luminal epithelial cell identity, antiandrogen sensitivity, and *CDK4*/*6* inhibitor resistance.

## 7. *ERG* and the Androgen Receptor (AR)

The androgen receptor (AR) is a transcription factor for a nuclear hormone receptor that regulates growth and development in normal prostate cells [46]. In normal cells, there is a certain combination of transcription factors and histone modifications that are specific to AR known as the AR cistrome. When this AR cistrome remains unchanged, it facilitates the normal development of prostate tissue. However, modifications to this cistrome have been implicated in enhancing tumor growth and progression in PCa [47]. Two factors that have been associated with reprogramming the AR cistrome include *FOXA1* and *HOXB13* [48]. *ERG* overexpression and *PTEN* loss have been associated with inducing the expression of these factors [49]. Furthermore, *ERG* can form a complex with AR through an AR-interacting motif (AIM), and a cysteine residue that is utilized to form cross-linkers between *AR* and *ERG* exists in this domain [46]. This allows it to act as a cofactor for AR to enhance DNA binding in both high- and low- affinity conditions [50]. Quantitatively, data have shown that the formation of this complex results in a three-fold increase in AR’s DNA-binding ability [46]. With this enhanced binding ability, AR sites have been observed to be concentrated at *FOXA1* and *HOXB13*, along with *AP1* and androgen response elements (ARE) [48]. Therefore, *ERG* overexpression results in AR site enrichment at the *FOXA1* and *HOXB13* motifs, resulting in increased expression. The increase in FOXA1 and HOXB13 is then found to modify the AR cistrome, further worsening tumorigenesis. *PTEN* loss also contributes to this, as PI3K and AR inhibition will be lost, resulting in constitutive AR and PI3K signalling [51]. *ERG* overexpression has been correlated to the relative resistance of AR-targeted therapeutics [52]. Furthermore, the overexpression of *ERG* in MSKPCa2 cells demonstrating high AR expression demonstrated increased cell growth in the presence of enzalutamide, which is a PCa treatment that works as an AR inhibitor [50]. Therefore, *ERG* and *AR* overexpression result in prostate cancer cells having increased tolerance to androgen receptor antagonists.

## 8. *ERG PTEN* and *TP53* Crosstalk

It is important to note that the overexpression of *ERG* via *TMPRSS2-ERG* fusion is not a definitive indicator of biochemical recurrence or survival and requires information regarding the *PTEN* and *TP53* status [44,53]. *PTEN* is a tumor suppressor, and its inactivation is one of the most significant prognostic biomarkers in prostate cancer [54]. Moreover, the *PI3K* pathway is directly controlled by *PTEN*, and *PTEN* loss has been correlated with a hyperactive *PI3K* signalling pathway, which regulates cell survival and proliferation, in various cancers [53]. Furthermore, the combination of *ERG* activation and *PTEN* or *TP53* loss can induce cell migration and transform prostatic intraepithelial neoplasia into invasive carcinoma [55].

## 9. *ERG* Stimulates Endothelial Gene Expression Utilizing the Canonical Wnt Signalling Pathway

*ERG* stimulates endothelial gene expression by utilizing the canonical Wnt signalling pathway. When Wnt ligands bind to frizzled receptors, it triggers a signalling cascade that prevents phosphorylation and therefore acts as a β-catenin stabilizer. The stability β-catenin allows it to translocate into the nucleus and promote the transcription of various other genes. Patients who have been identified as having *TMPRSS2-ERG* gene fusion have been shown to exhibit increased ERG activity, which has been directly correlated to mRNA levels with various Wnt ligands, including *WNT2*, *WNT3A*, and *WNT11* [56]. Frizzled receptors such as *FZD4*, which has been found to mediate EMT in prostate cancer cells [57,58], have been found to also be upregulated by ERG and to contribute to increased Wnt signalling activity (Figure 2).

FZD4 is known as an Wnt receptor in the Wnt signalling pathway and is directly related to the B-catenin signalling pathway [58]. *FZD4* is required for the oncogenic effects of ERG fusion. Interestingly, *FZD4* knockdown induces phenotype results in active B-integrin and E-cadherin expression that are similar to those of ERG knockdown. On the other hand, *FZD4* overexpression clearly reverses the impact of ERG knockdown in PCa cell lines, showing the direct correlation and co-regulation of these two genes.

In the Wnt signaling pathway, the binding of Wnts to various receptors results in the accumulation of ß-catenin, which, when recruited by the *LEF1* transcription factor, targets multiple oncogenes such as *c-MYC* and *MMPs* [59]. Studies have shown that the *LEF1* promoter is the most important *TMPRSS2*-*ERG* target of the Wnt signaling pathway, as *LEF1* is knockdown results in the complete inhibition of the *ERG*-induced Wnt pathway [59].

Another study also showed that Wnt signalling is involved in the self-renewal and differentiation of cancer stem cells. Moreover, in LNCap and C4-2B PCa cells, WNT3A levels have been positively correlated with prostasphere size and self-renewal, with increased expression resulting in a 1.5× increase in sphere formations [60]. Additionally, *WNT3A* has been shown to support the progression of PIN lesions into more advanced adenocarcinomas and to increase cell resistance to androgen deprivation [17]. Previous studies have shown that the increased expression of *WNT1* and *LEF1,* especially through the Wnt signaling pathway, is correlated with lethality, hormone resistance PCa, and the EMT pathway [61].

SFRP proteins are a group composed of five proteins that are structurally and functionally similar to Wnt ligands and that act as Wnt pathway antagonists by binding to Wnt proteins or Frizzled receptors [62]. There have been previous studies that present the low expression of SFRP1 proteins in PCa; however, the role of SFRPs in PCa remain controversial [62,63,64]. In VCap cells, SFRP1 appears to promote the transcriptional activity of AR and to cause an increase in the expression of *TMPRSS2-ERG*. This co-expression results in the increased migration and invasion of tumor xerographs [62].

## 10. Integrin Linked Kinase Pathway

Previous studies have demonstrated that ERG expression results in the upregulation of the Integrin-linked kinase (*ILK*), which further activates Snail via poly (ADP ribose) polymerase-1 (PARP-1), resulting results in the downregulation of E-cadherin and the activation of the EMT pathway [65,66]. Furthermore, *ILK* activation due to *ERG* activation results in the suppression of cytokeratin 8/18 and increased expression of N-cadherin and vimentin, which further induces EMT pathway activity [65]. *ILK* has been shown to be involved in the regulation of the EMT and proliferation pathways (Figure 2), and it has been shown to have increased expression in various cancers [65,66].

## 11. *ERG* Overexpression Upregulates PI3KB-PKB/Akt Pathway

*ERG* overexpression is often accompanied by the loss of *PTEN* in PCa, which further upregulates the Akt pathway due to a lack of inhibition [67]. The upregulation of the *AKT* signalling pathways has been observed to work in tandem with downstream *ERG* targets to make cancer progression more aggressive [68]. *PTEN* is a known inhibitor of AKT and acts by converting PIP_3_ to PIP_2_, directly opposing *AKT* signalling [69]. The AKT pathway is a key regulatory step in controlling the transcription of many genes that induce cell proliferation, metabolism, anti-apoptosis, genomic instability, and differentiation. In hindsight, the binding of a substrate leads to the activation of *PI3K*, resulting in the catalytic conversion of PIP_2_ to PIP_3_ [70]. Akt then interacts with PIP_3_ at the plasma membrane, which allows PDK1 to phosphorylate *Thr308*; this allows for the partial activation of AKT, which leads to the activation of *mTORC1* and tuberous sclerosis protein 2 (*TSC2*) [71]. The downstream effect of this partial activation promotes cellular proliferation and protein synthesis [72]. The full activation of *AKT* results in phosphorylation occurring both in the cytoplasm and in the nuclei of proteins such as *FOXO,* which leads to attenuation of apoptosis and the promotion of the proliferation, angiogenesis, and survival of cells [73].

## 12. Metalloproteinase MMP1,3,9 and ADAMTS1 Pathway

*ERG* has been shown to be recruited by the *Fos/Jun* complex, bind to the promoter, and activate matrix metalloproteinase 1 (MMP1) transcription. Furthermore, *ETS2* has been shown to directly activate both MMP1 and MMP3 [74,75]. Overall, *ERG* and *ETS2* induce the loss of focal adhesion and therefore the de-differentiation of the prostate cells [21]. Additionally, MMP9 has been shown to be positively correlated with *TMPRSS2-ERG,* which is upregulated in VCaP cells; however, the mechanism behind this interaction has yet to be determined [76]. MMP1 and MMP3 are proteolytic enzymes that degrade bonds and connective tissue in the basement membrane; therefore, their activation is directly correlated with the ability of cells to migrate and invade [21].

Another family of proteins that share the metalloproteinase domain with MMPs are disintegrin and metalloproteinase (*ADAMs*). They regulate various cell functions such as migration and invasion and have been known as cancer progression regulators. ERG has been shown to upregulate *ADAMTS1* genes in prostate cancer cells, which show an attraction towards fibroblasts [77].

ERG activates C-MYC to promote the de-differentiation of the PCa cells, and C-MYC has been identified as a major transcriptional factor that is involved in various cellular functions such as proliferation, differentiation, apoptosis, and cellular motility. *C-MYC* has also been shown to be a major oncogene that is implicated in the progression of prostate cancer [78]. It has been reported that C-MYC regulates and increases the expression of *AR* genes and helps to stabilize various proteins such as AR-FL and AR-V, which are observed in castration-resistant prostate cancer [79]. *C-MYC* knockdown in enzalutamide resistant PCa cells results in enhanced cells sensitivity to enzalutamide [79]. Previous studies conclude that C-MYC expression may be used as a predictor for biochemical recurrence in primary prostate tumors. C-MYC expression may not be directly related to ERG protein expression, but it is strongly correlated with TMPRSS2-ERG status [80]. It appears that the high expression of ERG due to *TMPRSS2-ERG* fusion results in the upregulation the *C-MYC* oncogene, which attenuates the differentiation of the prostate epithelium [38]. When ERG and C-MYC levels were reduced using siRNA, the upregulation of prostate differentiation genes such as *PSA*, *SL34A3/prostein,* and *MSMB* was observed [79]. Furthermore, studies have shown that when *ERG* is repressed, the C-MYC targets that are activated express self-renewal genes, resulting in decreased HSC differentiation and an increased HSC population [81]. Overall, it appears that ERG activates *C-MYC* in an effort to promote the de-differentiation of the prostate epithelial cells [21,82].

## 13. *ERG* and microRNAs

Kim et al. reported that in PCa, ERG directly binds to the ETS motif within the promoter of miR-200c and inhibits its expression [83]. miR-200c is one of the members of the miR-200 family that has been shown to be downregulated in metastatic compared to primary tumors, and its loss has been correlated with poor cell differentiation [21]. It was reported that miR-200 acts a tumor suppressor and downregulates EMT markers such as *ZEB1* and Vimentin [83]. Another study indicated that miR-221 is actually downregulated in patients with more aggressive PCa with *TMPRSS2-ERG* gene fusion [84]. miR-221 has been known to be overexpressed in various tumors such as bladder [85], glioblastoma [86], breast [87], and chronic lymphocytic leukemia [88]. It appears that in these types of cancers, miR-221 and miR-449 suppress the expression of p27 and p21, respectively, which are tumor suppressors, and are responsible for regulating the phase transition from G0 to S phase, which further stimulates proliferation [84,89]. ERG overexpression ultimately results in the overexpression of genes in the EMT pathway and enhances the migration and invasion ability of PCa cells [83]. Studies have shown that there is a correlation between the downregulation of miR-221 and metastatic tumors, but to this date. the mechanism of action has yet to be elucidated by researchers.

## 14. Nitric Oxide NO-cGMP Signaling Pathway

Previous studies have shown that the alpha1 and beta1 subunits of Soluble guanylyl cylase (sGC) and cGMP synthesis are elevated by *TMPRSS2-ERG* in PCa cells [90]. More importantly, studies have shown that not only does sGC inhibitor treatment suppress tumor growth and proliferation in *TMPRSS2-ERG*-positive PCa xerographs, but that it can also act strategically with an AR antagonist such as enzalutamide [90]. sGC is a regulator of the nitric oxide (NO)-cGMP signaling pathway. Upon the binding of NO, sGC synthesizes cGMP and activates protein kinase G (PKG), which is directly related to cell proliferation and tumorigenesis in various cancers [90,91,92].

## 15. TMPRSS2-ERG Fusion Upregulates CXCR4 Enhances Tumor Adhesion and Aggregation

Previous studies have concluded that there are eight ERG/Ets factor binding sites near the promoter of chemokine receptor type 4 (*CXCR4)* (Figure 3) and that *TMPRSS2-ERG* expression enhances the function and expression of *CXCR4* [93,94]. CXCL12 is a known CXCR4 receptor ligand, and the interactions between the two functions enhance tumor aggressiveness and increase the ability of cancer cells to adhere to the extracellular matrix [95,96]. The *CXCR4/CXCL12* axis has been shown to increase MMP expression, which promotes cell migration and growth in PCa [93]. An analysis of PCa demographics revealed relatively higher CXCR4 expression during PCa progression and in metastatic bone tissue compared to benign PCa cells that were reported. In vitro, a direct correlation between *TMPRSS-ERG* and CXCR4 was observed when ERG was knocked down [96].

## 16. TMPRSS2-ERG Binds to EMT Key Regulators *ZEB1/ZEB2*

Prior studies show that *TMPRSS2-ERG* can directly bind to the Zinc finger e-box binding homeobox 1 (*ZEB1*) promoter and binds to *ZEB2* modulators such as ILIR2 and SPINT1 and increases their overall expression [97]. *ZEB1* and *ZEB2* are members of the *ZEB* family of transcriptional factors and are key regulators in the EMT and disease progression pathways [98]. Furthermore, *ZEB1* knockdown revealed a significant decline in the migration and invasion capacity of *TMPRSS2-ERG*-expressing cells.

## 17. EZH2 Enhances ERG Oncogenic Activity

It appears that the enhancer of zeste jomolog 2 (*EZH2*), which is a histone H3K27 methyltransferase, catalyzes the methylation of *ERG* at the lysine 263 residue. This interaction promotes the translocation and DNA binding of *ERG* in the nucleus, which results in the enhancement of *the* oncogenic activity of ERG [99,100]. Studies have shown that the methylation of *ERG* at the lysine362 residue is associated with metastatic properties and increased tumorigenic characteristics in cell lines [99]. Further, these interactions seem to enhance the progression of PCa from non-invasive lesions to invasive adenocarcinomas in Erg/Pten mice [99].

## 18. *ERG* and Tight Junction Protein CLDN5

Recent Studies have shown that ERG is positively correlated with the expression of an important tight junction protein known as Claudin 5 (CLDN5); it has been observed that there are two major ERG binding sites near the promoter of the *CLDN5* gene, making ERG a direct transcriptional regulator of *CLDN5*. Moreover, CLDN5 is known as a member of the 24 tetraspan transmembrane protein family and one of the major components of the tight junction strands that are responsible for regulating barrier functions [101]. This protein is involved in many of the processes related to vascular homeostasis. Research has shown that the knockout of ERG expression in mice resulted in increased endothelial cell permeability due to decreased CLDN5 expression [102]. Interestingly, *CLDN5* upregulation results in decreased the cell migration and invasion ability of lung cancer cells due to the decreased permeability of the cell membrane due to the enhancement of the CLDN5 tight junctions [103,104]. In addition to ERG’s function in maintaining vascular homeostasis, it also plays a critical role in maintaining the population of hematopoietic stem cells (HSC) by regulating their differentiation. A recent study demonstrated that *ERG* is necessary for arresting HSCs in a dormant G_0_ phase by analyzing the number of cells within each phase of the cell cycle. *CLDN5* has been implicated as a negative regulator of many biological processes such as angiogenesis, cell migration, and vascular permeability; alternatively, it has also been shown to be a positive regulator of tight junction assembly, cell population differentiation, protein binding, and endothelial barrier development [103,105,106]. Furthermore, *TNF*-alpha has been classified as a direct inhibitor of *ERG* and *CLDN5* by extension [101].

## 19. *ERG* Upregulates Distal-Less Homeobox-1(DLX1)

It is thought that *ERG* upregulates distal-less homeobox 1 (*DLX1*) by interacting with enhanced bound *AR* and *FOXA1* [107]. To support this hypothesis, it has been observed that when *ERG* and therefore *DLX1* transcription is inhibited via BET inhibitors, there is a significant reduction in the oncogenic effects of *DLX1* [107]. *DLX* is a transcriptional factor and is part of the homeobox-containing family [108]. Several malignancies, such as those including the prostate, have been linked to the deregulation of the homeobox gene, and therefore, *DLX1* has been validated as a potential PCa biomarker [107,109].

## 20. GSK3B and WEE1 Induces TMPRSS2-ERG Degradation

Hong et al. [110] revealed that the glycogen synthase kinase 3 beta (*GSK3B*) and WEE1 induce *TMPRSS2-ERG* degradation via the dual phosphorylation of ERG threonine-187 and tyrosine-190. Such phosphorylation allows for the recognition and degradation of the *ERG* oncoprotein by the E3 ubiquitin ligase FBW7. *GSK3B* and *WEE1* have been found to be associated with the DNA damage that is induced proteasomal degradation in PCa [111]. The relationship between *TMPRSS2-ERG* and *PTEN* has been described previously, but it is interesting to note that this degradation pathway is eradicated in the case of *PTEN* loss or *GSK3B* inactivation. This has further been implicated with the growth of chemoresistant PCa cell lines in culture and in mice.

## 21. Summary

Clinical data demonstrate the incidence of *ERG* overexpression in approximately 50% of all patients diagnosed for prostate cancer. The most common mechanism associated with *ERG* overexpression arises from the gene fusion of *TMPRSS2* and *ERG*. This fusion allows for the androgen responsive promoter *TMPRSS2* to act on *ERG*, resulting in its transcriptional upregulation. Researchers have begun to elucidate the role of *ERG* in the oncogenic mechanisms associated with tumorigenesis and cancer progression in their search for potential non-invasive biomarkers and novel targeted therapeutics. Here, we reviewed multiple pathways that are affected by the upregulation of *ERG* in PCa. We conclude that *ERG* is a pivotal mediator in tumor development and is involved in many cellular processes, including cell proliferation, de-differentiation, angiogenesis, and cancer stem cell homeostasis and is implicated in the epithelial to mesenchymal transition pathway. Future research should explore the potential applications of *ERG* in improving diagnostic and prognostic methods in PCa as well as potential opportunities for therapeutic targeting.

## Figures and Tables

**Figure 1 ijms-23-04772-f001:**
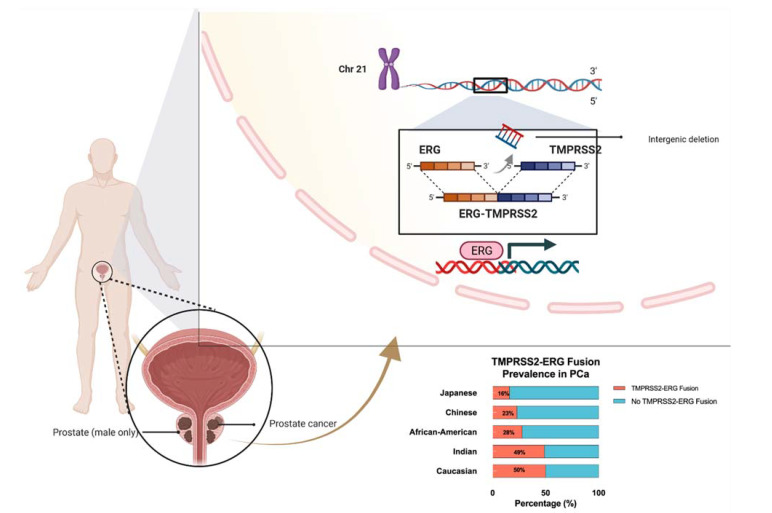
Schematic diagram of *TMPRSS2-ERG* gene fusion on chromosome 21 and the ethnic prevalence of *TMPRSS2-ERG* gene fusion in prostate cancer patients.

**Figure 2 ijms-23-04772-f002:**
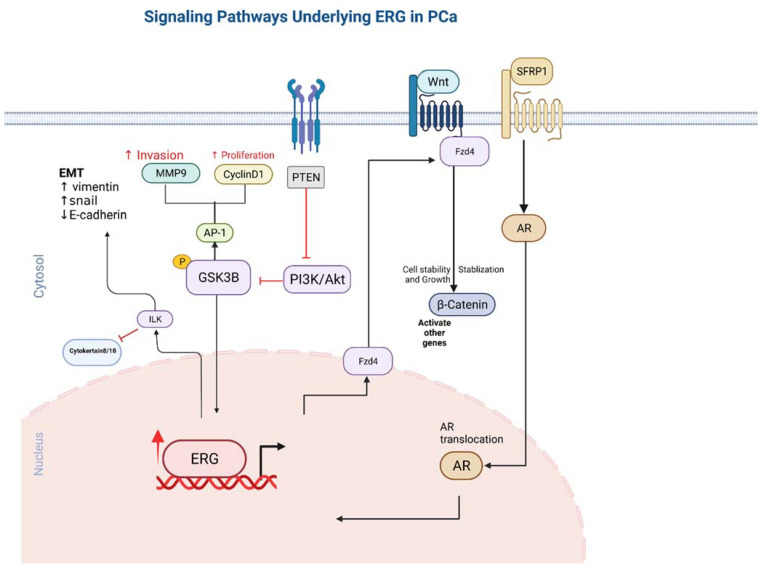
The pathways involved in ERG overexpression and related outcomes in tumor cells. Directional arrows indicate activation.

**Figure 3 ijms-23-04772-f003:**
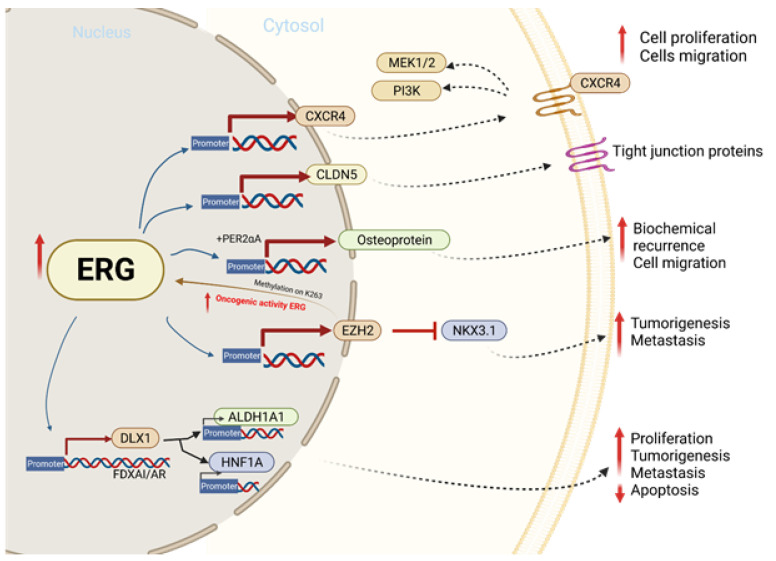
The molecular mechanism involved in ERG overexpression and related outcomes in tumor cell proliferation, migration, invasion, and metastasis.

## Data Availability

Not applicable.

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
