# Peer review of "The Expression of Proto-Oncogene ETS-Related Gene (ERG) Plays a Central Role in the Oncogenic Mechanism Involved in the Development and Progression of Prostate Cancer"

_ijms, 2022, doi:10.3390/ijms23094772_

Round 1
Reviewer 1 Report
This study was reported the role of ERG and the fusion of the ERG and TMPRSS2 genes for prostate cancer. The reviewer would like to suggest some critiques as follows.
- On line 6, “and 4*” is correct? Who?
- On line 111, the authors spelled out TMPRSS 2. However, TMPRSS2 was not spelled out on line 19 and 47. Which is correct? Regarding PTEN on line 163 and 184, which is correct?
- On line 20, the authors described that approximately 85% of patients with prostate cancer have the fusion of the ERG and TMPRSS2 genes. Conversely, the TMPRSS2-ERG fusion appears in appears in approximately 50% of peripheral zone vs 12% of transition zone. On line 390, the incidence of ERG overexpression in 50% of patients with prostate cancer. These are true?
- On line 37, what is “the most common type of cancer”? High morbidity? Incidence?
- On line 49, “That being said,….transition zone tumours.” Is unclear. The authors should revise this sentence.
- What is the primary endpoint in this study? The reviewer can not understand the aim of this study.
Author Response
Reviewer 1
- On line 6, “and 4*” is correct? Who?
# Thank you. This has been fixed.
- On line 111, the authors spelled out TMPRSS 2. However, TMPRSS2 was not spelled out on line 19 and 47. Which is correct? Regarding PTEN on line 163 and 184, which is correct?
#Thank you for your feedback. We have added the full name of TMPRSS2 and PTEN at the very first encounter through the manuscript per your suggestion.
- On line 20, the authors described that approximately 85% of patients with prostate cancer have the fusion of the ERG and TMPRSS2 genes. Conversely, the TMPRSS2-ERG fusion appears in appears in approximately 50% of peripheral zone vs 12% of transition zone. On line 390, the incidence of ERG overexpression in 50% of patients with prostate cancer. These are true?
# In line 20, it is mentioned that the ERG-TMPRSS2 gene fusion is reported in 85% of all gene fusions in prostate cancer and not in 85% of prostate cancer patients.
- On line 37, what is “the most common type of cancer”? High morbidity? Incidence?
#New references have been added to demonstrate the number of new cases and death due to PCa.
- On line 49, “That being said,….transition zone tumours.” Is unclear. The authors should revise this sentence.
#This sentence has been revised and clarified.
- What is the primary endpoint in this study? The reviewer can not understand the aim of this study.
# Thank you. Our review dedicated to summarizing the various features of TMPRSS2-ERG fusion in PCa, the primary aim is to gather and summarize all the current data and discoveries in a short and concise manner.
Reviewer 2 Report
Authors review the role of the proto-oncogene ERG in the prostate cancer. Although the paper is very interesting, the data presented are confusing and not related to each other. The review results a collection of pieces that seem disconnected and this makes it difficult to read and understand the text. In some cases, the same sentence is repeated several times. The references are too old, out of 105 citations, only 15 are from the last 4 years. In my opinion, the paper needs greater organization and above all greater data integration.

Author Response
Reviewer 2
Authors review the role of the proto-oncogene ERG in the prostate cancer. Although the paper is very interesting, the data presented are confusing and not related to each other. The review results a collection of pieces that seem disconnected, and this makes it difficult to read and understand the text. In some cases, the same sentence is repeated several times. The references are too old, out of 105 citations, only 15 are from the last 4 years. In my opinion, the paper needs greater organization and above all greater data integration.
# Thank you for your feedback. The references been updated. Overall revised and organized accordingly.
Round 2
Reviewer 1 Report
The authors revised this review according to the reviewer’s recommendation. The reviewer believes that this paper will provide useful information for the readers.